# Investigation of the Changes in Concentrations of Vitamin D-Binding Protein and Lactoferin in Plasma and Peritoneal Fluid of Patients with Endometriosis

**DOI:** 10.3390/ijms24097828

**Published:** 2023-04-25

**Authors:** Barbara Lisowska-Myjak, Ewa Skarżyńska, Monika Wróbel, Grzegorz Mańka, Mariusz Kiecka, Michał Lipa, Damian Warzecha, Robert Spaczyński, Piotr Piekarski, Beata Banaszewska, Artur Jakimiuk, Tadeusz Issat, Wojciech Rokita, Jakub Młodawski, Maria Szubert, Piotr Sieroszewski, Grzegorz Raba, Kamil Szczupak, Tomasz Kluz, Marek Kluza, Mirosław Wielgoś, Piotr Laudański

**Affiliations:** 1Department of Biochemistry and Phatmacogenomics, Medical University of Warsaw, 02-097 Warsaw, Poland; 2Department of Laboratory Medicine, Medical University of Warsaw, 02-097 Warsaw, Poland; 31st Department of Obstetrics and Gynecology, Medical University of Warsaw, 02-015 Warsaw, Poland; 4Angelius Provita Hospital, 40-611 Katowice, Poland; 5City South Hospital Warsaw, 02-781 Warsaw, Poland; 6OVIklinika Infertility Center, 01-377 Warsaw, Poland; 7Center for Gynecology, Obstetrics and Infertility Treatment Pastelova, 60-198 Poznan, Poland; 8Division of Infertility and Reproductive Endocrinology, Department of Gynecology, Obstetrics and Gynecological Oncology, Poznan University of Medical Sciences, 61-701 Poznan, Poland; 9Chair and Department of Laboratory Diagnostics, Poznan University of Medical Sciences, 61-701 Poznan, Poland; 10Department of Reproductive Health, Insitute of Mother and Child in Warsaw, 01-211 Warsaw, Poland; 11Department of Obstetrics and Gynecology, Central Clinical Hospital of the Ministry of Interior, 02-507 Warsaw, Poland; 12Department of Obstetrics and Gynecology, Insitute of Mother and Child in Warsaw, 01-211 Warsaw, Poland; 13Collegium Medicum, Jan Kochanowski University in Kielce, 25-369 Kielce, Poland; 14Clinic of Obstetrics and Gynecology, Provincial Combined Hospital in Kielce, 25-736 Kielce, Poland; 15Department of Gynecology and Obstetrics, Medical University of Lodz, 90-419 Lodz, Poland; 16Department of Surgical Gynecology and Oncology, Medical University of Lodz, 90-419 Lodz, Poland; 17Department of Fetal Medicine and Gynecology, Medical University of Lodz, 90-419 Lodz, Poland; 18Clinic of Obstetric and Gynecology in Przemysl, 37-700 Przemysl, Poland; 19Department of Obstetrics and Gynecology, University of Rzeszow, 35-310 Rzeszow, Poland; 20Department of Gynecology, Gynecology Oncology and Obstetrics, Institute of Medical Sciences, Medical College of Rzeszow University, 35-310 Rzeszow, Poland; 21Premium Medical Clinic, 04-359 Warsaw, Poland; 22Lazarski University, 02-662 Warsaw, Poland; 23Department of Obstetrics, Gynecology and Gynecological Oncology, Medical University of Warsaw, 02-091 Warsaw, Poland; 24Women’s Health Research Institute, Calisia University, 62-800 Kalisz, Poland

**Keywords:** endometriosis, vitamin D-binding protein, lactoferrin, peritoneal fluid

## Abstract

An evaluation of the association between the concentrations of vitamin D-binding protein and lactoferrin in the plasma and peritoneal fluid may facilitate the elucidation of molecular mechanisms in endometriosis. Vitamin D-binding protein and lactoferrin concentrations were measured by ELISA in plasma and peritoneal fluid samples from 95 women with suspected endometriosis as classified by laparoscopy into groups with (n = 59) and without endometriosis (n = 36). There were no differences (*p* > 0.05) in the plasma and peritoneal fluid concentrations of vitamin D-binding protein and lactoferrin between women with and without endometriosis. In women with endometriosis, there was a significant correlation between plasma and peritoneal fluid vitamin D-binding protein concentrations (r = 0.821; *p* = 0.000), but there was no correlation between lactoferrin concentrations in those compartments (r = 0.049; *p* > 0.05). Furthermore, in endometriosis, lactoferrin was found to correlate poorly with vitamin D-binding protein (r= −0.236; *p* > 0.05) in plasma, while in the peritoneal fluid, the correlation between those proteins was significant (r = 0.399; *p* = 0.002). The characteristic properties of vitamin D-binding protein and lactoferrin and the associations between their plasma and peritoneal fluid concentrations found in women with endometriosis may provide a novel panel of markers to identify high-risk patients in need of further diagnostic measures.

## 1. Introduction

To date, no non-invasive or minimally invasive tests have been developed to specifically identify individuals at high-risk of actual disease among women with suspected endometriosis. Laparoscopy and positive histology remain the gold standard for the definitive diagnosis of endometriosis [1,2,3]. Due to the non-specific symptoms of the disease and invasive character of surgery, the diagnostic delay may exceed seven years [4]. The measurement of markers in the body fluids, including plasma and peritoneal fluid (PF), could theoretically serve as an alternative approach to the preliminary diagnosis of the condition [2,5]. However, the pathogenesis of endometriosis remains unclear, and relatively little is known about the molecular basis of metabolic processes involved in the development and activity of the disease, which makes it difficult to find markers which are both sensitive and specific [3,6]. A rational solution to facilitate the understanding of systemic and local biological processes associated with the development of endometriosis could be the selection of specific proteins detected in the plasma and PF which work in tandem to form a panel to fulfil their biological functions in different fluid compartments.

Endometriosis is a chronic inflammatory disease associated with disturbances in the levels of many different active molecules in the eutopic endometrium, peritoneal fluid and plasma, such as those of matrix metalloproteinases, adhesive molecules, cathepsins, chemokines, autoantibodies and other immunologically related substances [7,8,9,10,11,12,13,14]. It is also linked with an increased release of biologically active compounds from erythrocytes into the peritoneal cavity [5,15,16,17,18]. Endometriosis is also a disease that has a negative impact not only on the physical health of affected patients, but also on their mental health. The disease’s symptoms and the mental state resulting from it are closely linked to each other in this condition [19]. In addition, the infertility often found in this population negatively affects the quality of life and mental state of the patients [20].

Another important aspect in the context of endometriosis is the costs directly and indirectly associated with the disease. These consist of both those related to medical care and those resulting from the patient’s loss of productivity. Surgical treatment turned out to be a significant expense [21], suggesting that advances in the diagnostic process and its possible independence from performing laparoscopy will result in a significant reduction in expenses.

Based on the properties of vitamin D-binding protein (VDBP) and lactoferrin (Lf), and their confirmed links with endometriosis, they were selected as potential and related biomarkers.

VDBP (58 kDa) is a glycosylated α_2_—globulin produced in the liver and located predominantly in the serum, a major plasma carrier protein of vitamin D metabolites to target organs, and involved in the transportation of fat and endotoxins. It is an important factor in the actin scavenging system released from damaged cells and enhances the monocyte and neutrophil chemotactic activity of C5-derived peptides. Following deglycosylation during inflammatory processes, VDBP becomes converted to the macrophage activating factor (MAF) [5,6,16]. Circulating levels of VDBP depend on the specific Gc polymorphism, nonspecific acute phase response and estrogen levels in women [22,23]. VDBP concentrations in ectopic endometrial tissue with endometriosis were significantly higher than those in normal endometrial tissue [6]. On the other hand, the PF levels of VDBP in women with endometriosis were lower, which was attributed to the increased conversion of VDBP to MAF [5].

Lf (78.0 kDa) is a single-chain glycoprotein found in human mucosal secretions and in milk, and is also stored in secondary neutrophil granules and released during degranulation. As an iron-binding glycoprotein, it plays a key role in the innate immunity response and has powerful anti-inflammatory properties as it prevents the formation of oxygen radicals. Extracellular Lf concentrations can be used as an index of polymorphonuclear neutrophil activation [15]. Lf levels in the PF are lower in minimal endometriosis compared to more advanced stages of the disease [15,18,24].

The aim of the study was the concurrent evaluation of VDBP and Lf concentrations in the plasma and peritoneal fluid of women with suspected endometriosis, diagnosed endometriosis and without endometriosis to elucidate the association(s) between the two proteins involved in the systemic and local pathological processes accompanying this disease.

## 2. Results

The differences in VDBP and Lf concentrations and their proportions (VDBP/Lf ratio) in the plasma and PF between the three groups of women with suspected endometriosis, without endometriosis and with diagnosed endometriosis are shown in Table 1.

The plasma and PF VDBP and Lf measurements and the ratios did not significantly differ between the three groups of women (*p* > 0.05). The VDBP concentrations in both the plasma and PF were over 1000-fold higher than the Lf concentrations. The VDBP and Lf measurements in the plasma and PF demonstrate high dispersion and approximately 100-fold coefficients of variation.

Table 2 presents the associations shown as coefficients of correlation between the plasma and PF measurements in the three groups of women.

There was a significant correlation (*p* < 0.05) of plasma VDBP with PF VDBP and PF Lf in each group. The endometriosis group differed from the two other groups in that there was an absence of any significant association (*p* > 0.05) between VDBP and Lf in the plasma in contrast to a significant correlation between VDBP and Lf in the PF and no significant association between Lf concentration in the plasma and in the PF.

The scatter plot in Figure 1 displays the measurements of VDBP and Lf in the plasma and PF of individual subjects with and without endometriosis.

Based on the measurements of the distribution of plasma VDBP and Lf displayed in Figure 1, women with endometriosis and without endometriosis were classified into two groups by the arbitrarily selected plasma VDBP cut-off value of 1000 µg/mL. The <1000 µg/mL groups included 30 women with endometriosis and 22 women without endometriosis, and the >1000 µg/mL groups included 29 women with endometriosis and 14 women without endometriosis.

Table 3 presents the classification of women with suspected endometriosis, without endometriosis and with diagnosed endometriosis into subgroups by the cut-off value of plasma VDBP of 1000 µg/mL. The other corresponding plasma and PF measurements are also included.

Higher plasma VDBP concentrations of >1000 µg/mL were associated with significantly increased VDBP concentrations in the PF in the three groups and increased Lf concentrations in the PF in the groups of suspected and diagnosed endometriosis.

The graph in Figure 2 evaluates the differences in the correlations between VDBP and Lf in the plasma and PF between women with endometriosis and without endometriosis. The vertical line separates the correlations between VDBP and Lf observed in the VDBP ranges of <1000 µg/mL and >1000 µg/mL. The coefficients of correlation of VDBP with Lf in the plasma and PF in women with endometriosis and without endometriosis classified by the cut-off value of plasma VDBP of 1000 µg/mL are presented in Table 4.

The associations between the VDBP and Lf concentrations in the plasma and PF shown in Figure 2 and Table 4 distinguish between women with diagnosed endometriosis and those without endometriosis. Characteristic findings of diagnoses of endometriosis included a negative correlation between VDBP and Lf concentrations in the plasma when the plasma VDBP was <1000 µg/mL and a positive correlation between VDBP and Lf concentrations in the PF when the plasma VDBP was >1000 µg/mL.

## 3. Discussion

The results demonstrate associations between the levels of VDBP and Lf in the plasma and peritoneal fluid which are specific for the diagnosis of endometriosis and potentially may facilitate the distinction between women with and without endometriosis.

An eager search for new biomarkers of endometriosis without the need for diagnostic surgery and histopathology has been prompted by the lack of sensitive screening tests for the non-invasive diagnosis of the condition [2,3]. Proteomic technologies have demonstrated that several individual proteins are differently expressed in various body fluids of patients with endometriosis versus those without this condition [2,5]. Investigations of links between proteins with different biological functions have failed to identify markers which would be sensitive and specific to the prompt diagnosis and assessment of disease activity [2,5]. The measurements of VDBP and Lf levels in the plasma and PF in the present study also do not reliably classify women with diagnosed endometriosis.

Since no individual protein has been found to correlate with disease activity or endometriosis symptoms, further research should focus on investigating panels of biomarkers. The aim is to understand how individual proteins with well-known independent activity and a molecular basis interact with each other and to investigate their possible role in the systemic and local pathology associated with endometriosis.

As already mentioned in the introduction, endometriosis is characterized by inflammation at the sites of abnormally located endometrium-like tissue and increased erythrocyte counts in the peritoneal cavity [5,15,17]. The confirmed independent biological properties of VDBP and Lf [5,15,22,23] suggest their involvement in the regulation of these processes. Measurements of plasma and peritoneal VDBP and Lf performed in parallel allowed an investigation of their interactions in the vascular system and the peritoneal cavity and of the differences in their activities in each of these compartments. A significant, positive correlation between the VDBP concentrations in the plasma and in the PF in both women with endometriosis and without endometriosis possibly indicate that this 58 kDa protein may easily penetrate from the vascular bed into the peritoneal cavity as its molecular weight is in the upper limit of deliverable molecules, which allows such penetration [22]. No correlation between the concentrations of Lf (with a molecular weight of 78.0 kDa) in the plasma and in the PF and no significant correlation between the plasma concentrations of VDBP and Lf were found between women with endometriosis and without endometriosis.

The question arises of whether or not any additional factors present in the vascular bed and the local microenvironment of endometriosis lesions modify the interaction between VDBP and Lf. Published experimental studies [15,22,25,26] strongly suggest that the involvement of VDBP and Lf in the regulation of neutrophil activity in both plasma and PF may suggest an important functional link between the two proteins. According to some authors [26], neutrophils are elevated in the systemic circulation and PF of endometriosis patients, but whether or not and how neutrophils actually contribute to endometriosis pathophysiology remains poorly understood. VDBP is able to bind to specific, activated regions on the neutrophil plasma membrane using myeloperoxidase released from the azurophilic granules [26]. VDBP shedding from neutrophils is mediated by intracellular elastase, whose inhibition by specific serine protease inhibitors may lead to VDBP accumulation on the neutrophil plasma membrane [25]. Lf as an anti-inflammatory protein located in specific granules in neutrophils is used as a marker for neutrophil activation [15,18,26]. In endometriosis, an overloading of the peritoneal milieu with elevated hemoglobin concentrations and highly toxic iron promotes a pro-oxidative/pro-inflammatory milieu which may enhance the release of iron-binding Lf from activated neutrophils [15].

VDBP is both a systemic parameter measured in the plasma and a specific parameter measured in a variety of biological fluids [22]. Considering its potential key role in the development of endometriosis, in the present study, the subjects with suspected endometriosis were subdivided using an arbitrary cut-off value for plasma VDBP into two groups: one with low (<1000 µg/mL) plasma levels and one with high (>1000 µg/mL) plasma levels of VDBP. Both groups included women who were later diagnosed with endometriosis and those without endometriosis. With lower plasma VDBP, a negative correlation between plasma VDBP and Lf was seen only in endometriosis women, while with higher plasma VDBP there was a positive correlation between peritoneal VDBP and Lf, again solely in women with endometriosis.

The negative correlation between VDBP and Lf at lower plasma VDBP concentrations may suggest the active role of VDBP in the regulation of systemic processes accompanying inflammation [5,27]. VDBP clears extracellular actin released from dead or damaged cells for removal from circulation, largely in the liver, while extracellular actin consumes VDBP and reduces its plasma concentration [22,25]. Decreased plasma concentrations of VDBP are a valuable marker of cell and tissue damage in acute tissue injury and they are accompanied by increases in the VDBP–actin complex formation due to the effect of VDBP conversion to MAF [22,23,25].

However, the role of VDBP in local inflammation, represented by neutrophil granulocyte activation, remains unclear. Based on the present findings, the low peritoneal concentrations of VDBP in women with endometriosis may be due to low concentrations of VDBP being transferred from the plasma to the peritoneal fluid and hence decreased neutrophil activation and degranulation in the PF. The proportional increases in the peritoneal concentrations of VDBP and Lf observed at higher plasma VDBP concentrations demonstrate the active involvement of VDBP and Lf in the regulation of the local intrauterine environment in endometriosis. This finding may confirm the observations made by other authors that neutrophil granulocytes in endometriosis patients might have a lower ability to respond to weak activation signals, but when more extensive endometriosis develops, stronger pro-inflammatory signals are present [18].

To date, there have been only a few clinical observations of low serum VDBP in women with infertility [28] and low peritoneal VDBP in infertile women with endometriosis [5]. The authors hypothesize that VDBP depletion in endometriosis lesions may result from VDBP contact with B and T cells and their ability to convert VDBP to MAF. Low peritoneal VDBP levels in women with endometriosis may indicate low neutrophil activation and confirm the observations made by other authors of lower peritoneal Lf levels in mild endometriosis vs. advanced stages of endometriosis [15]. The parallel high peritoneal VDBP and Lf concentrations detected in this study give rise to the question of whether or not neutrophil degranulation and the release of Lf from neutrophil granules, though having potentially anti-inflammatory effects, may actually pose a threat to local tissue damage in the microenvironment of endometriosis. Previous studies [24] based on parallel Lf determination in plasma and PF performed on the same clinical material confirm the individual regulation of Lf concentrations in PF. The small number of women with endometriosis and the lack of adequate numbers for the creation of statistical groups characterizing subsequent stages of endometriosis are undoubtedly a weakness of the results presented in this work. Further high-quality research is needed using specific parameters of neutrophil function to elucidate the role of neutrophils in the development and activity of endometriosis

A strength of the paper is its innovative nature, which clearly outlines directions for further research. In addition, it is worth noting the carefully collected material, especially the peritoneal fluid, with particular attention to the purity of the samples taken. A weakness is the small number of patients, so that the results obtained can only be regarded as indicators of the direction of further research and not as findings with definite clinical utility.

In summary, the observation that VDBP and Lf work in tandem in the plasma and peritoneal fluid suggests their potential use in non-invasively identifying women with and without endometriosis among patients suspected of having the condition. To confirm the correlation of VDBP with Lf in the plasma and peritoneal fluid that would serve as a marker for endometriosis, it is necessary to rigorously establish the plasma ranges of VDBP. A negative correlation of VDBP with Lf at low plasma VDBP concentrations suggests that the two proteins are involved in the systemic regulation of endometriosis-associated inflammation. Proportional increases in peritoneal VDBP and Lf concentrations at high plasma VDBP concentrations suggest the biological role of the two in local endometriosis lesions.

## 4. Materials and Methods

### 4.1. Subjects

The study was conducted on 95 women aged 19 to 43 years (mean age: 31.5 ± 5.2 years) presenting symptoms which required clinical evaluation for endometriosis. All subjects underwent a gynecological examination and a transvaginal ultrasound followed by diagnostic laparoscopy, which included careful inspection of the uterus, fallopian tubes, ovaries, pouch of Douglas, and pelvic peritoneum and was performed by experienced gynecologists. Diagnostic laparoscopy confirmed the diagnosis of endometriosis in 59 women and an absence of the condition in 39 women.

A detailed description of the patient recruitment as well as the inclusion and exclusion criteria are presented in article entitled “Autoantibody screening of plasma and peritoneal fluid of patients with endometriosis” [29]. Briefly, the exclusion criteria included a non-typical menstrual cycle (<25 days or >35 days), and age upon inclusion in the study of below 18 years or above 40 years, any type of hormone therapy in the preceding three months, diagnosed or suspected malignancy, a history of or present pelvic inflammatory disease, uterine fibroids, polycystic ovaries or a possible autoimmune disorder. The group selection process is shown in the attached flowchart [29].

All individual participants included in the study were requested to fill out the World Endometriosis Research Foundation (WERF) clinical questionnaire, and written informed consent was obtained from each participant. The study was approved by the Ethics Committee at the Medical University of Warsaw (KB/223/2017).

### 4.2. Material

Plasma was obtained after the centrifugation of blood collected from the antecubital vein into an EDTA tube prior to a laparoscopy procedure.

Peritoneal fluid (PF) was obtained by aspiration using a Veress needle under direct visualization, immediately upon the insertion of the laparoscope to avoid contamination with blood.

Plasma was centrifuged at 2500× *g* and PF at 1000× *g*, both for 10 min at 4 °C, transferred in portions into smaller test tubes and stored at −80 °C until further assays.

### 4.3. Laboratory Methods

Concentrations of Lf and VDBP were measured by the enzyme-linked immunosorbent assay (ELISA) using the commercially available AssayMax^TM^ Human Lactoferrin ELISA Kit (AssayproP LLC—St. Charles, MO, USA) and the Human (DBP) ELISA kit, SunRed (Shanghai, China). The ELISA tests were performed in duplicate according to the manufacturer’s instructions, and concentrations were expressed in µg/mL.

### 4.4. Statistical Analysis

Statistical analyses were performed for the three study groups: women referred for a clinical evaluation of suspected endometriosis (n = 95), women without endometriosis (excluded by clinical evaluation) (n = 36) and women diagnosed with endometriosis (n = 59). For each group, the Shapiro–Wilk test was performed to assess the normality of the distribution of the results. The concentrations of VDBP and Lf in the plasma and PF are presented as mean ± standard deviation (
$\bar{x}$ ± SD), range, median, and coefficient of variation (CV%). The Mann–Whitney U test was used to compare differences between the independent groups, and the Wilcoxon test was used to compare pairs of observations. The correlations between the parameters were assessed by the Spearman’s rank order correlation test, which gave statistically significant correlation coefficients (R_s_). The statistical analyses were performed using STATISTICA version 13.3., www.statsoft.com, accessed on 1 January 2019. A *p*-value of <0.05 was considered statistically significant.

## Figures and Tables

**Figure 1 ijms-24-07828-f001:**
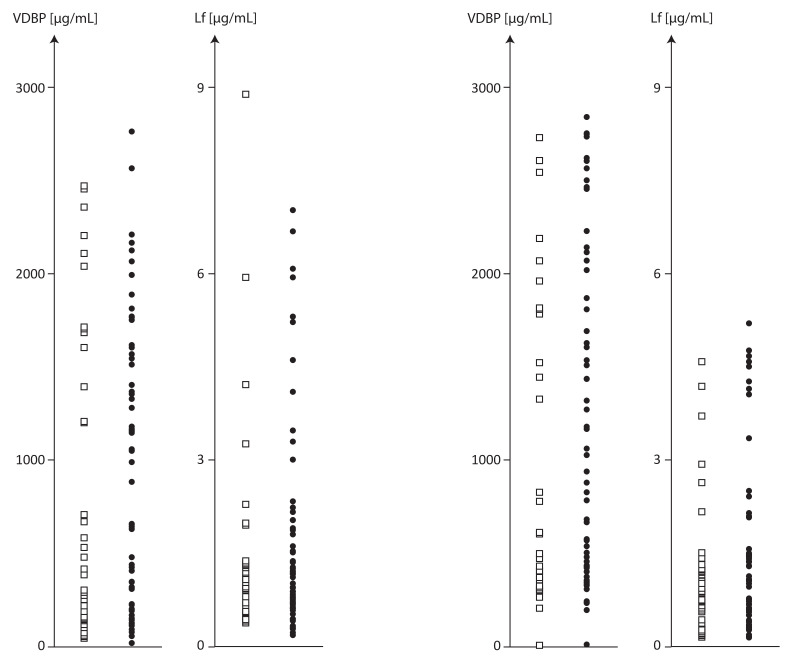
Plasma VDBP and Lf concentrations and PF of subjects with and without endometriosis.

**Figure 2 ijms-24-07828-f002:**
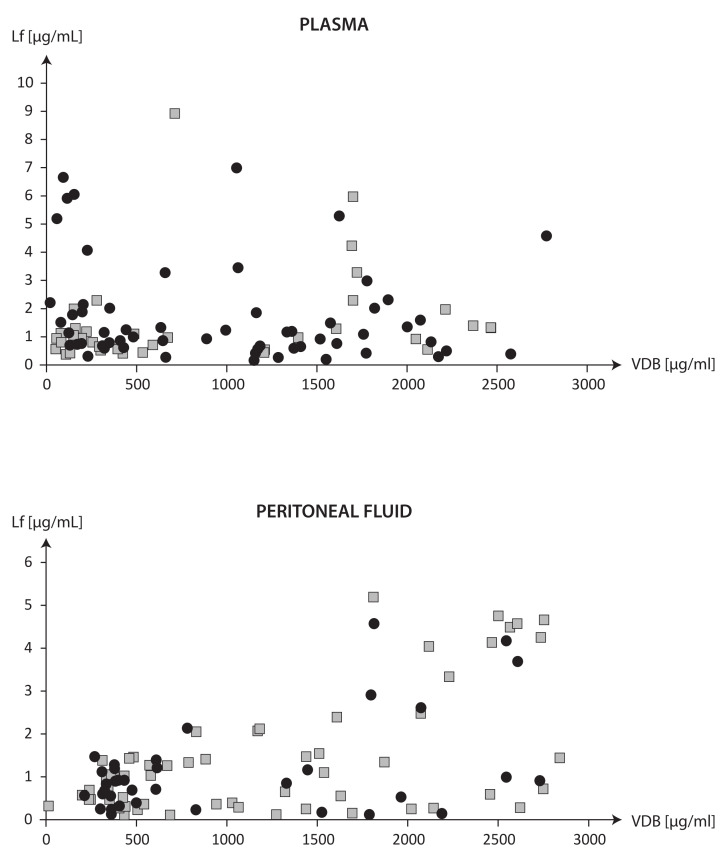
The correlations between VDBP and Lf in the plasma and PF between women with endometriosis and without endometriosis.

**Table 1 ijms-24-07828-t001:** VDBP and LF concentrations and their proportions (VDBP/LF ratio) in plasma and peritoneal fluid in women with suspected endometriosis, without endometriosis and with confirmed endometriosis.

Parameter Mean ± SD Median (Dispersion) CV % *	Suspected Endometriosis N = 95		No Endometriosis N = 36		Endometriosis N = 59		Differences between Groups *p* **	
	Plasma	Peritoneal Fluid	Plasma	Peritoneal Fluid	Plasma	Peritoneal Fluid	Plasma	Peritoneal Fluid
**VDBP µg/mL**	945.8 ± 779.3 **661.8** (20.1–2773.9) 82.4%	1138.0 ± 856.5 **787.1** (14.5–2839.8) 75.3%	895.6 ± 832.0 **510.3** (51.4–2467.5) 92.9%	1007. ± 833.4 **552.8** (212.4–2730.2) 82.7%	976.4 ± 751.0 **993.9** (20.1–2774.0) 77.0%	1217.7± 867.6 **1028.6** (14.5–2839.8) 71.3%	0.814	0.417
**LF µg/mL**	1.66 ± 1.72 **1.05** (0.16–8.92) 103.8%	1.31 ± 1.31 **0.91** (0.11–5.19) 99.8%	1.51 ± 1.71 **0.97** (0.37–8.92) 113.0%	1.15 ± 1.12 **0.87** (0.12–4.57) 97.1%	1.74 ± 1.73 **1.14** (0.16–6.99) 99.5%	1.41 ± 1.41 **0.98** (0.11–5.19) 100.3%	0.865	0.806
**VDBP/LF ratio**	1201.7 ± 1638.4 **596.1** (9.1–7682.2) 136.3%	1997.6 ± 3048.98 **647.2** (45.3–15,401.7) 152.6%	845.8 ± 895.2 **488.6** (58.0–3867.1) 105.8%	2057.5 ± 3639.6 **665.9** (183.0–15,401.7) 176.9%	1418.8 ± 1934.0 **630.8** (9.1–7682.2) 136.3%	1961.1 ± 2658.6 **643.6** (45.3–10,991.5) 135.6%	0.722	0.886

* coefficient of variation (CV %), ** *p* values for multiple comparisons, Kruskal–Wallis test. The median is shown in bold.

**Table 2 ijms-24-07828-t002:** Correlation coefficients (Spearman) between VDBP and LF concentrations in plasma and peritoneal fluid (PF) in women with suspected endometriosis, without endometriosis and with confirmed endometriosis.

Correlations	Study Groups		
	Suspected Endometriosis (n = 95)	No Endometriosis (n = 36)	Endometriosis (n = 59)
**Plasma VDBP vs. Plasma LF**	−0.004, *p* = 0.968	**0.374, *p* = 0.025**	−0.236, *p* = 0.072
**Plasma VDBP vs. PF VDBP**	**0.819, *p* = 0.000**	**0.793, *p* = 0.000**	**0.821, *p* = 0.000**
**Plasma VDBP vs. PF LF**	**0.401, *p* = 0.000**	**0.405, *p* = 0.014**	**0.413, *p* = 0.001**
**PF VDBP vs. PF LF**	**0.349, *p* = 0.001**	0.256, *p* = 0.132	**0.399, *p* = 0.002**
**Plasma LF vs. PF LF**	0.184, *p* = 0.074	**0.376, *p* = 0.024**	0.049, *p* = 0.714

Bold font indicates significant (*p* < 0.05) correlations between VDBP and LF.

**Table 3 ijms-24-07828-t003:** Effect of an increase in plasma VDBP concentration of >1000 µg/mL on plasma and peritoneal fluid parameter concentrations in women with suspected endometriosis, without endometriosis and with confirmed endometriosis.

Parameter Mean ± SD Dispersion	Symptomatic Endometriosis N = 95		*p*	No Endometriosis N = 36		*p*	Endometriosis *N =* 59		*p*
	VDBP < 1000 (N = 52)	VDBP > 1000 (N = 43)		VDBP < 1000 (N = 22)	VDBP > 1000 (N = 14)		VDBP < 1000 (N = 30)	VDBP > 1000 (N = 29)	
**Plasma**									
**VDBP µg/mL**	314.5 ± 229.9 20.1–993.9	1709.2 ± 452.6 1055.4–2773.9	0.000	287.6 ± 203.6 51.4–712.2	1851.1 ± 433.4 1207.5–2467.5	0.000	334.2 ± 249.0 20.1–993.9	1640.7 ± 452.9 1055.4–2773.8	0.000
**LF µg/mL**	1.65 ± 1.81 0.26–8.92	1.66 ± 1.63 0.16–6.99	0.952	1.27 ± 1.77 0.37–8.92	1.89 ± 1.59 0.44–5.97	0.048	1.93 ± 1.81 0.27–6.65	1.55 ± 1.66 0.16–6.99	0.179
**Ratio VDBP/LF**	379.1 ± 421.9 9.1–2469.5	2196.4 ± 1985.2 151.1–7682.2	0.000	369.8 ± 337.5 58.0–1211.7	1593.7 ± 993.9 1207.4–2467.5	<0.001	385.9 ± 480.0 9.1–2469.5	2487.3 ± 2276.7 151.1–7682.2	0.000
**Peritoneal fluid**									
**VDBP µg/mL**	468.7 ± 246.1 14.5–1437.8	1947.5 ± 585.5 781.5–2839.8	0.000	415.6 ± 141.7 245.9–2752.6	1937.5 ± 559.6 781.5–2730.2	0.000	507.6 ± 297.1 14.5–1437.8	1952.3 ± 607.2 829.5–2839.8	0.000
**LF µg/mL**	0.74 ± 0.43 0.11–1.47	2.0 ± 1.65 0.12–5.19	<0.001	0.75 ± 0,40 0.13–1.47	1.78 ± 1.56 0.12–4.57	0.139	0.73 ± 0.45 0.11–1.47	2.11 ± 1.71 0.12–5.19	0.003
**Ratio VDBP/LF**	1050.6 ± 1200.0 45.3 –6230.2	3142.9 ± 4074.7 348.7–15,401.7	<0.001	862.4 ± 850.0 183.0–3570.8	3935.4 ±5318.0 367.1–15,401.7	0.013	1188.5 ± 1400.5 45.3–6230.2	2760.3 ± 3362.8 348.7–10,991.5	0.008

**Table 4 ijms-24-07828-t004:** Effect of an increase in plasma VDBP concentration of >1000 µg/mL on the interrelationship between VDBP and LF concentrations in plasma and peritoneal fluid.

Study Groups	Correlations between VDBP and LF Concentrations	
	Plasma VDBP µg/mL	
	<1000 µg/mL (n = 52)	>1000 µg/mL (n = 43)
	Plasma	
**No endometriosis**	R = 0.062, *p* = 0.785	R = 0.290, *p* = 0.314
**Endometriosis**	**R = –0.400, *p* = 0.029**	R = 0.05, *p* = 0.794
	**Peritoneal fluid**	
**No endometriosis**	R = 0.159, *p* = 0.468	R = 0.289, *p* = 0.338
**Endometriosis**	R = 0.231, *p* = 0.329	**R = 0.416, *p* = 0.022**

## Data Availability

The data presented in this study are available on request from the corresponding author. The data are not publicly available due to privacy.

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
