# Peer review of "Investigation of the Changes in Concentrations of Vitamin D-Binding Protein and Lactoferin in Plasma and Peritoneal Fluid of Patients with Endometriosis"

_ijms, 2023, doi:10.3390/ijms24097828_

Round 1
Reviewer 1 Report
The manuscript "Plasma and peritoneal vitamin D-binding protein and lactoferrin as a specific test panel for the diagnosis of endometriosis" is an interesting manuscript on the role of vitamin D-binding protein and lactoferrin as a specific test panel for the diagnosis of endometriosis. The work is original and well-structured, giving novelty to scientific literature. The design of the project is appropriate, but the number of patients considered for this study is low (this could be a potential bias for the results of the study), so it needs to be evaluated by a statistician. The English language is acceptable. In the structure of the manuscript the materials and methods have to be added after the introduction instead after the conclusion, please revise this aspect.
Author Response
Answer to reviewer comment:
Thank you very much for your valuable comment. Of course, materials and methods should be added after the introduction, this has already been corrected in the manuscript. We also agree that the number of patients is not large and certainly no definitive conclusions can be drawn from it. However, we would like to point out that the study is innovative, the assessment of vitamin D binding protein and lactoferrin concentrations in plasma and peritoneal fluid, to our knowledge, has not been performed before in patients with endometriosis. We consider our results as a beginning for further research, outlining the direction of possible relationships. In addition, our work is part of a project to evaluate potential biomarkers of endometriosis, the funding for which allowed us to assemble the presented study group. The results of other assays on a similar number of patients have been previously published, for example in PMID: 36675136, 36141853, 36289716, 36555313 and 36749097.
Reviewer 2 Report
Endometriosis poses an ever growing pathology in gynecology. Studies that are trying to approach endometriosis and uncover diagnostic correlations are of paramount importance and I fully appreciate the work of the authors.
I recommend a small change in line 86 regarding English language, 'released' instead of 'releasing'.
Also I recommend changing the title to better correlate with the results as it seams that advanced disease endometriosis is more suited for VDBP and Lf studies.
I think that further studies on the VDBP cutoff would be of real value.
Author Response
Answer to reviewer comment:
Thank you very much for your favourable feedback and valuable comments. The change in line 86 has already been made to the manuscript.
We have also changed the title to more adequately reflect the outcome of our work and to be consistent with the conclusions reached.
Obviously, the determination of the cut-off value is very important and would be a potentially valuable parameter applicable in clinical practice. This will be the aim of our next study, in which we plan to broaden the study group so that the results obtained are as reliable as possible.
Reviewer 3 Report
Thank you for the opportunity to review the interesting study incorporating one of the novel panel of biomarkers for endometriosis diagnostics.
However there are several issues that has to be addressed:
1) Materials and Methods has to be sectioned before Results and Discussion.
2) Introduction has to be expanded with 1-2 sentence regarding ameliorating quality of life in this population of patients (please cite relevant, recently published articles - PMID: 34718292 and PMID: 35819491). Furthermore, expand the Introduction and Discussion section with more data about possible pathogenetic pathways in endometriosis (PMID: 36142815).
3) Please include strengths and limitations of your study.
4) Please emphasize the potential clinical implication regarding the public health resources - cost/benefit of this approach?
5) Please provide the study flowchart in Materials and Methods section.
Author Response
Answer to reviewer comment:
Thank you very much for your pertinent comments. Material and Methods has indeed been moved to the right place.
The quality of life and psychological state of patients with endometriosis is obviously a very important aspect worth mentioning . After reviewing the publications, we have added a section in the introduction outlining this issue. Similarly, in the case of costs associated with endometriosis- based on the cited publication, the above issue has been addressed in the introduction. This is undoubtedly an important issue justifying the value of conducting further research into improving the diagnostic process for this disease.
Regarding the presumed pathogenesis of endometriosis, an immune factor is extensively studied, as we mentioned in the introduction of our article. The reported publication on soluble CTLA4 antigens is a further confirmation of this thesis and is therefore included in the cited articles.
Obviously, stating the strengths and weaknesses of the work presented is important for the correct interpretation of the work, and thus this issue has been added to the manuscript.
In accordance with a very pertinent comment, a study flowchart summarising the group selection criteria has been added to the manuscript. Hopefully, this will make the group selection process clearer to the reader.
Round 2
Reviewer 3 Report
N/A